# Tolerance to ambiguous uncertainty predicts prosocial behavior

Marc-Lluís Vives[1] & Oriel FeldmanHall[2]

Uncertainty is a fundamental feature of human life that can be fractioned into two distinct psychological constructs: risk (known probabilistic outcomes) and ambiguity (unknown probabilistic outcomes). Although risk and ambiguity are known to powerfully bias nonsocial decision-making, their influence on prosocial behavior remains largely unexplored. Here we show that ambiguity attitudes, but not risk attitudes, predict prosocial behavior: the greater an individual's ambiguity tolerance, the more they engage in costly prosocial behaviors, both during decisions to cooperate (experiments 1 and 3) and choices to trust (experiment 2). Once the ambiguity associated with another's actions is sufficiently resolved, this relationship between ambiguity tolerance and prosocial choice is eliminated (experiment 3). Taken together, these results provide converging evidence that attitudes toward ambiguity are a robust predictor of one's willingness to engage in costly social behavior, which suggests a mechanism for the underlying motivations of prosocial action.

---

[1] Center for Brain and Cognition, Universitat Pompeu Fabra, 25-27 Ramon Trias Fargas St, Barcelona 08005, Spain. [2] Department of Cognitive, Linguistic, and Psychological Sciences, Brown University, 190 Thayer St, Providence, RI 02912, USA. Correspondence and requests for materials should be addressed to O.F. (email: feldmanhall@brown.edu)

Humans live in a complex, dynamic world characterized by ubiquitous uncertainty. In most social interactions, an individual must assess whether engaging with another person will reap rewarding or punishing consequences. For example, understanding whether a new colleague can be trusted with confidential information or will be a cooperative team player on an upcoming project could be critical to one's job success. Most social decisions, including deciding whether to trust or cooperate with others, are inherently uncertain[1–3] and demand a constant estimation of the possible risks associated with each option under consideration. Despite this, very little is understood about how uncertainty influences how people perceive and act during dyadic social interactions.

Within the nonsocial decision-making literature, uncertainty is known to be a potent influencer of how people assess the value of available options[4,5]. Individuals exhibit a routine aversion to uncertainty, choosing a safe option that results in a small, reliable payout over an uncertain option that can yield a high but unreliable payout[6]. These decisions under uncertainty have been further fractionated into decisions under risk, where probabilities are known[7], and decisions under ambiguity, where probabilities are unknown[8–10]. Research illustrates that attitudes toward risk and ambiguity are separate psychological constructs with little overlap between the two[11,12]. It is now also well-documented that individuals are far more averse to ambiguous compared to risky uncertainty, consistently avoiding outcomes that are associated with unknown probabilistic outcomes[13–15]. Accordingly, ambiguity is thought to be a more profound form of uncertainty with a stronger impact on behavior[8,16,17].

In social contexts, these uncertainty considerations become especially acute when one must decide how to engage with another person[18,19]. Many prosocial choices require an individual to initially place their own well-being into the hands of another, which if reciprocated, can facilitate mutual gain for all involved parties[20]. Yet, the act of placing one's own welfare into another's hands can be characterized by a loss of control and thus an increase in uncertainty. Prior theoretical work has capitalized on this idea, suggesting that social decisions are inherently risky[21,22]. For example, choosing to trust can effectively be construed as a perceived vulnerability stemming from uncertainty about another's motives and possible actions[23], as trusting another can expose an individual to a possible moral hazard if the trust is not repaid. According to this theory, individuals faced with social uncertainty should hold beliefs that are represented with a known (e.g., risky) probability distribution.

Empirical research, however, has failed to find a relationship between attitudes toward risk and prosocial behavior, such as decisions to trust[24,25]. This may be because assessing whether an individual is trustworthy—or cooperative, generous, or kind—is more analogous to estimating unknown probabilistic outcomes, as it is rare to know with probabilistic certainty how another's actions will unfold[26,27]. In other words, it becomes difficult to estimate another's behavior when intentions and motives are hidden. This inability to apply known probabilities to a set of outcomes further affects the actions an individual might subsequently take, rendering social exchanges rife with ambiguous uncertainty. For these reasons, many social choices are likely to be defined by ambiguous, rather than risky, uncertainty[28]. And yet, research exploring the effects of uncertainty on social decision-making has focused on the psychological construct of risk, while failing to consider the dimension of ambiguity. Thus, one critical question that has received little attention is how ambiguity attitudes shape prosocial decision-making.

One plausible account is that individuals who are ambiguity-tolerant will be more likely to engage in highly uncertain prosocial behaviors, such as deciding to trust or cooperate with a stranger. This is supported by literature within the nonsocial domain, which illustrates that individuals with higher ambiguity tolerance are more optimistic about outcomes occurring in one's favor[29,30]. Effectively, this account would suggest that those who are ambiguity optimistic would expect strangers to repay trust or reciprocate cooperative actions, whereas individuals who are ambiguity pessimistic would assume that strangers cannot be trusted.

Here we investigate this possibility, testing whether ambiguity—but not risk—attitudes better predict decisions to cooperate and trust in a series of dyadic games. If ambiguity is endemic to social decision-making, then individuals who are ambiguity-seeking (i.e., tolerant) should exhibit greater prosocial behavioral patterns when engaging with others. Across multiple experiments, we formally estimate risk and ambiguity attitudes in subjects while they play a gambling task known to successfully capture both types of uncertainty[11,31]. On every trial, subjects make choices between a certain, safe monetary option and an uncertain monetary option that either has known probabilistic outcomes (i.e. risk) or unknown probabilistic outcomes (i.e. ambiguity). Subjects' risk and ambiguity attitudes are then computed using a maxmin utility model[32], where $\alpha$ indicates the subject's risk sensitivity and $\beta$ indicates the subject's ambiguity sensitivity (see Methods for modeling details). In order to assess whether these risk or ambiguity attitudes predict prosocial behavior, subjects then play a public goods game (PGG, experiment 1), a trust game (TG, experiment 2) or a prisoner's dilemma (PD, experiment 3). Thus, across all experiments, subjects are given the option to either select a prosocial choice that could potentially enhance everyone's payout, or to defect, which would curtail any uncertainty associated with the social exchange. Results support the hypothesis that the degree to which an individual cooperates or trusts is linked to their attitudes toward ambiguity, but not risk. We find that those who are more tolerant to ambiguous uncertainty engage in greater cooperative and trusting behavior, and that resolving the ambiguity associated with another's behavioral repertoire abolishes this relationship.

## Results

**Cooperation under uncertainty.** In experiment 1, subjects ($N = 103$) first played a gambling task (Fig. 1a), where on each trial they decided between a sure payout of $5 or the option to play the lottery. Each lottery varied in terms of the amount of risk (25, 50, and 75% of winning the money), ambiguity (24, 50, and 74%), and potential payoffs (from $5 to $125). For example, in a risky trial, subjects could choose between a sure outcome and a gamble with a 50% chance of winning $20. These probabilities were denoted by a picture of a blue and a red bar that corresponded to an actual bag filled with 100 blue and red chips (placed beside the subject in the testing room). In an ambiguous trial, subjects were presented with a similarly colored bar; however, a proportion of the bar was occluded, leaving subjects partially informed of the composition of the chips.

After completing the gambling task, subjects played the PGG (Fig. 1b), in which they decided whether to invest their full $8 endowment (renewed on every trial) into a common pool shared by three other players (i.e., defined as decisions to cooperate). The money invested by all players was multiplied by two and then redistributed across all four players. For instance, if the subject was the only player to invest her $8 and the other three players kept their money (i.e., defined as decisions to freeride), then the subject, and every other player, would end up with a payout of $4 for that round. If all players invested their money, then each player would end up with a payout of $16 for that round. Alternatively, subjects could decide to freeride and not contribute

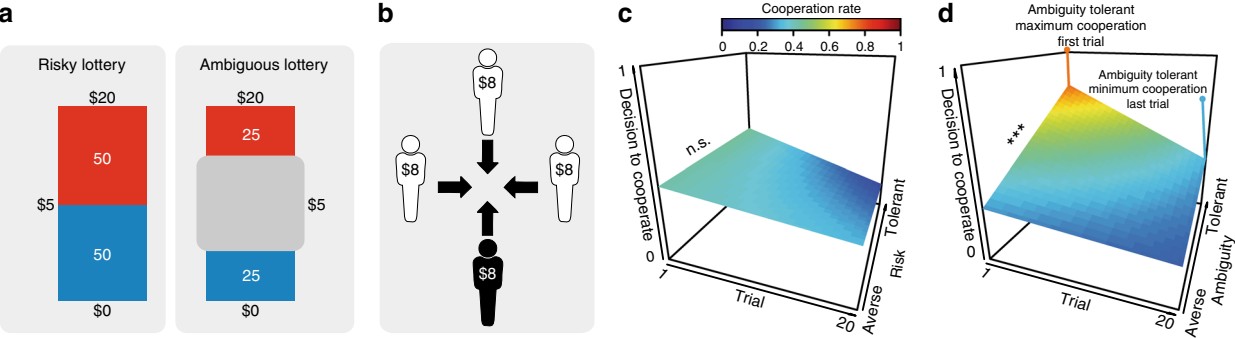

**Fig. 1** Experiment 1 task structure and results. Subjects first completed a computerized gambling task, which consisted of bags filled with 100 red and blue poker chips—denoted by a red and a blue bar. **a** The left panel represents a risky lottery with 50% chance of winning $20, while the right panel is an example of an ambiguous lottery where 50% of the chips are occluded. Subjects could always choose between playing the lottery or getting a sure payout of $5. **b** In the PGG, each player was endowed with $8 and had to decide to contribute to a common pool or keep the entire $8 for themselves. Money contributed to the pool was doubled and split equally amongst all players. **c** A regression reveals risk attitudes (even when accounting for feedback presented over the task) do not modulate decisions to cooperate. **d** In contrast, ambiguity-tolerant individuals are more willing to cooperate with others, and this relationship is modulated by receiving feedback over time. The surfaces represent regression planes that were calculated by extracting the predicted values from each regression, where 1 indicates full cooperation and 0 indicates full defection. Asterisks indicate significant differences (***$p <$ 0.001)

to the pool, thus keeping their $8 and earning additional payoffs if other players contributed to the pool. After making each decision, subjects were presented with the decisions of the other players (different players were presented on each round) and the resulting payout for everyone (i.e., feedback). The frequency at which the other players contributed to the pool was approximately 60% across the task.

Overall, subjects cooperated 35% of the time, illustrating a bias toward freeriding rather than cooperating. To test the hypothesis that cooperation rates are linked to attitudes toward ambiguity and not risk, we ran a trial-by-trial hierarchical logistic regression modeling subjects' cooperative behavior in the PGG as a function of trial number (to account for receiving feedback over time, which inherently reduces ambiguity), and their individual risk ($\alpha$) and ambiguity ($\beta$) attitudes. As predicted, individuals who were ambiguity-tolerant (i.e., exhibited a greater tolerance toward uncertainty in the first gambling task; denoted by $\beta > 1$) were more likely to cooperate (Fig. 1d; Table 1). We found no such relationship with risk attitudes (Fig. 1c; Table 1; see Supplementary Fig. 2 for a representation of raw behavioral data). Testing for the significance between these two predictors revealed that the coefficient for ambiguity was significantly different from risk ($z$ (102) = 2.14, $p = 0.03$; coefficients taken from the simple regressions reported in Supplementary Tables 5 and 6), suggesting that ambiguity attitudes have a unique and distinctive effect on prosocial behavior independent from risk attitudes.

Interestingly, modeling trial number revealed that as the task progressed and subjects gained more knowledge about what is typically contributed to the common pool, ambiguity attitudes became less predictive of cooperative behavior (Fig. 1d; Table 1). This indicates that as subjects become increasingly informed of how other people behave in the PGG, the degree of uncertainty about the other players is attenuated, which tautologically reduces the level of ambiguity associated with cooperating.

These findings are the first that we are aware of to illustrate link between ambiguity tolerance and decisions to engage in prosocial behavior, suggesting that the primary type of uncertainty that influences social choice is ambiguity and not risk. However, subjects in experiment 1 were faced with a binary decision to either contribute money toward a common good or selfishly keep the money. This type of decision space is limited and may fail to capture the more granular type of choices made in the real world[33], such as the amount of resources one is willing to allocate

**Table 1 Experiment 1: Risk and ambiguity attitudes in the Public Goods Game**

| DV | Coefficient ($\beta$) | Estimate (SE) | t-value | P-value |
|---|---|---|---|---|
| Cooperation | Intercept | −0.15 (0.19) | −0.75 | 0.45 |
| | Risk attitude | 0.20 (0.20) | 1.01 | 0.31 |
| | Trial | −0.08 (0.01) | −6.61 | <0.001*** |
| | Risk attitude × trial | −0.02 (0.01) | −1.68 | 0.09† |
| | Ambiguity attitude | 0.72 (0.22) | 3.31 | <0.001*** |
| | Ambiguity attitude × trial | −0.03 (0.01) | −2.20 | 0.03* |

Cooperation$_{i,t}$ = $\beta_0$ + $\beta_1$ risk attitude$_i$ × $\beta_2$ trial number$_{i,t}$ + $\beta_3$ ambiguity attitude$_i$ × $\beta_4$ trial number$_{i,t}$ + $\varepsilon$
Where risk and ambiguity attitudes are indexed by subject ($i$) and trial number is indexed by subject and trial ($i$, $t$). Ambiguity attitudes were inverted to align on the same scale as risk attitudes, and risk and ambiguity attitudes were standardized before being entered into the regression. Cooperation is coded as defect (0) and cooperate (1); AIC = 2064
†$p < 0.1$; *$p < 0.05$; ***$p < 0.001$

to the public good, rather than one's overall willingness to cooperate. Accordingly, we wanted to examine whether the effect of ambiguity tolerance and its relationship with enhanced prosocial behavior exists when the decision space is more reflective of the continuous nature of our everyday decisions.

In addition, a core feature of ambiguous uncertainty is that it can be resolved—or partially resolved—by gathering information about the state of the decision space[34]. This can be achieved in social contexts by gossiping with others[35], vicariously observing how others behave[36], or actively gathering information by directly engaging with others. To assess whether the relationship between ambiguity attitudes and prosocial behavior is mediated by the amount of knowledge one has about the decision space, we manipulated the degree to which subjects could obtain more information about their partners. For example, in experiment 1, subjects could gather information about how people generally behave in the PGG without cooperating themselves (i.e., freeriding). Effectively, subjects could test whether people on the whole cooperate or take advantage of the ability to freeride. Yet, in many social situations, behaving prosocially is a sine qua non condition for gathering information about other individuals within the environment. In other words, vicariously learning

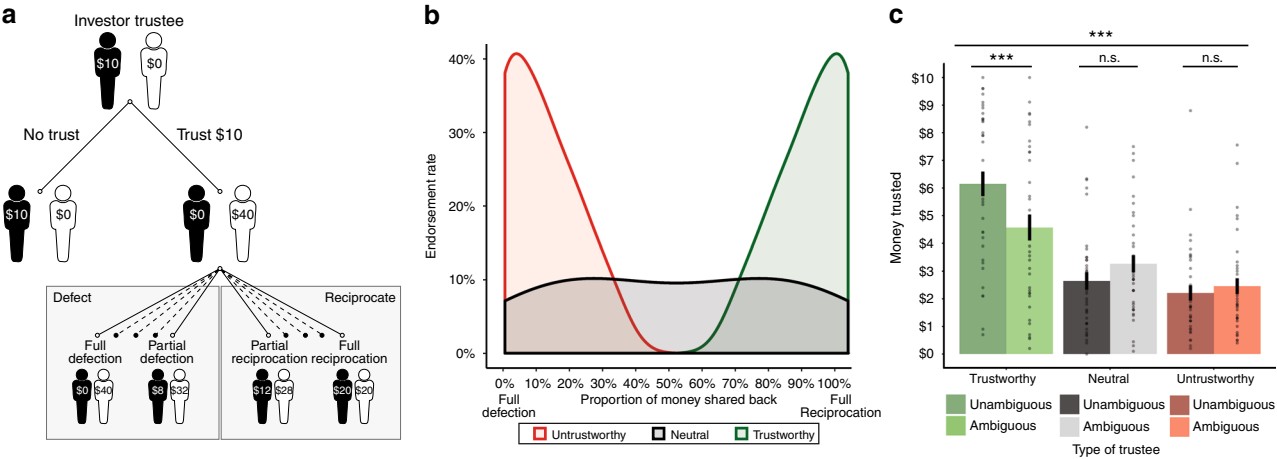

**Fig. 2** Experiment 2 task structure and results. **a** In the trust game, subjects (investors) were endowed with $10 and decided whether to share their money with the trustee, knowing that the money entrusted was quadrupled once sent to the trustee. If subjects entrusted their money, the trustee could reciprocate any proportion of money from returning nothing ($0) to the maximum amount (i.e., double the invested money). The decision tree depicts an example of a decision to trust and the various possibilities the trustee could choose from, from full defection to full reciprocation. **b** Each trustee was yoked to a different reciprocation algorithm: trustworthy trustees typically reciprocated, the neutral trustees were equally likely to defect or reciprocate (e.g., a uniform density across all possibilities); untrustworthy trustees typically defected. The proportion of money shared back refers to the maximum reciprocation (i.e., double the invested money. **c** A repeated measures ANOVA reveals subjects were sensitive to the trustworthiness of the trustee, entrusting the most money to trustworthy trustees and the least money to untrustworthy trustees. This effect was modulated by ambiguity levels associated with the trustees (whether feedback was presented), such that greater ambiguity had a negative effect for the trustworthy trustees, and no effect for neutral and untrustworthy trustees. Asterisks indicate significant differences (\*\*\**p* < .001). Error bars represent ± 1 standard error of the mean

about others' behavioral patterns without actively engaging with them is rare. Thus, rather than allowing subjects to obtain non-costly information about the other players, in experiment 2, we yoked information gathering and the resolution of ambiguity to actively making costly prosocial choices.

**Trust under uncertainty.** Experiment 2 examined the relationship between ambiguity attitudes and prosocial behavior in the domain of trust. Subjects ($N = 37$) first played the gambling task described in experiment 1, in which we estimated subjects' risk ($\alpha$) and ambiguity ($\beta$) attitudes (Fig. 1a), before partaking in a modified iterative TG. In our modified TG (Fig. 2a), subjects acted as the investor, entrusting their $10 endowment (renewed on every trial) with six different trustees. Subjects were told that any amount they shared (in $1 increments) would be quadrupled and sent to the trustee. The trustee could then decide to reciprocate and share back any proportion of the money (up to half of the total amount received). For example, if the subject decided to share the full $10, the trustee could share back any amount between $0 (i.e., full defection) and $20 (i.e., full reciprocation; for more details, see Methods). Unbeknownst to the subject, each trustee's "trustworthiness" was manipulated by systematically varying the amount of money reciprocated, such that there were three discrete types of players: a trustworthy trustee who often shared back most of the money; an untrustworthy trustee who rarely reciprocated; and a neutral trustee, who reciprocated 50% of the time such that he had a flat distribution of how much money was returned (Fig. 2b, see Methods and Supplementary Methods for details).

In addition, in order to exogenously manipulate ambiguity levels, we modified the amount of feedback subjects had access to. For example, while one trustworthy player was associated with full feedback (i.e., feedback was presented on all trials), the other trustworthy player was associated with partial feedback, such that feedback about whether the player reciprocated or defected was presented on only half of the trials. This resulted in a three

(players' trustworthiness) by two (ambiguity level) design, such that there were six types of trustees encountered across the task. Together, this task design allowed us to test whether the relationship between ambiguity attitudes and prosocial behavior (1) is observable across multiple social contexts, (2) exists in the face of costly prosocial behavior, and (3) is sensitive to the granularity of continuous choices.

Dovetailing with previous work[37], subjects were successfully able to distinguish between the three types of trustees (repeated measures analysis of variance (ANOVA): $F(2,72) = 57.37$, $p < 0.001$, $\eta^2 = 0.61$; Fig. 2c): subjects entrusted the most money to the trustworthy trustees (mean money trusted = $5.36, SD ± 2.59), and the least amount of money to the untrustworthy trustees ($M = $2.33$, SD ± 1.56). Moreover, this effect was influenced by the amount of ambiguity associated with each trustee (revealed by an interaction between type of trustee and the exogenous ambiguity manipulation; repeated measures ANOVA: $F(2,72) = 19.62$, $p < 0.001$, $\eta^2 = 0.35$). Whereas partial feedback (ambiguity) had a negative effect for the trustworthy trustees (ambiguous trustworthy trustee: $M = $4.57$, SD ± 2.84; unambiguous trustworthy trustee: $M = $6.15$, SD ± 2.69; paired *t*-test: $t(36) = -4.96$, $p < 0.001$; comparisons were Bonferroni-corrected), there was no effect for the untrustworthy trustees (ambiguous untrustworthy trustee: $M = $2.46$, SD ± 1.73; unambiguous untrustworthy trustee: $M = $2.21$, SD ± 1.67; paired *t*-test: $t(36) = -1.11$, $p = 0.28$), nor for the neutral trustees (ambiguous neutral trustee: $M = $3.26$, SD ± 1.90; unambiguous neutral trustee: $M = $2.65$, SD ± 1.91; paired *t*-test: $t(36) = 2.24$, $p = 0.03$, which did not survive Bonferroni correction). These findings suggest that in order to trust, subjects require more knowledge and less ambiguity about whether the trustee was indeed trustworthy. In contrast, for the untrustworthy trustees, the amount of ambiguity had little effect on whether subjects entrusted their money: any signal that a player is untrustworthy, regardless of whether it is ambiguous or not, is enough to dampen one's willingness to trust.

**Table 2 Experiment 2: Ambiguity attitudes in the Trust Game**

| DV | Coefficient (β) | Estimate (SE) | t-value | P-value |
|---|---|---|---|---|
| Money trusted | Intercept | 0.91 (0.11) | 8.48 | <0.001*** |
| | Ambiguity attitude | −0.03 (0.11) | −0.32 | 0.75 |
| | Untrustworthy | −0.26 (0.06) | −4.50 | <0.001*** |
| | Trustworthy | 0.63 (0.09) | 6.94 | <0.001*** |
| | Ambiguity attitude × untrustworthy | 0.13 (0.05) | 2.31 | 0.02* |
| | Ambiguity attitude × trustworthy | 0.09 (0.09) | 0.98 | 0.33 |

Money trusted$_{i,t}$ = $\beta_0$ + $\beta_1$ ambiguity attitude$_i$ × $\beta_2$ type of trustee$_{i,t}$ + $\varepsilon$
Where ambiguity attitude is indexed by subject ($i$) and type of trustee is a categorical variable, such that the neutral trustee serves as the reference category. Ambiguity attitudes were inverted and standardized before being entered into the regression; AIC = 10,492
*$p < 0.05$; ***$p < 0.001$

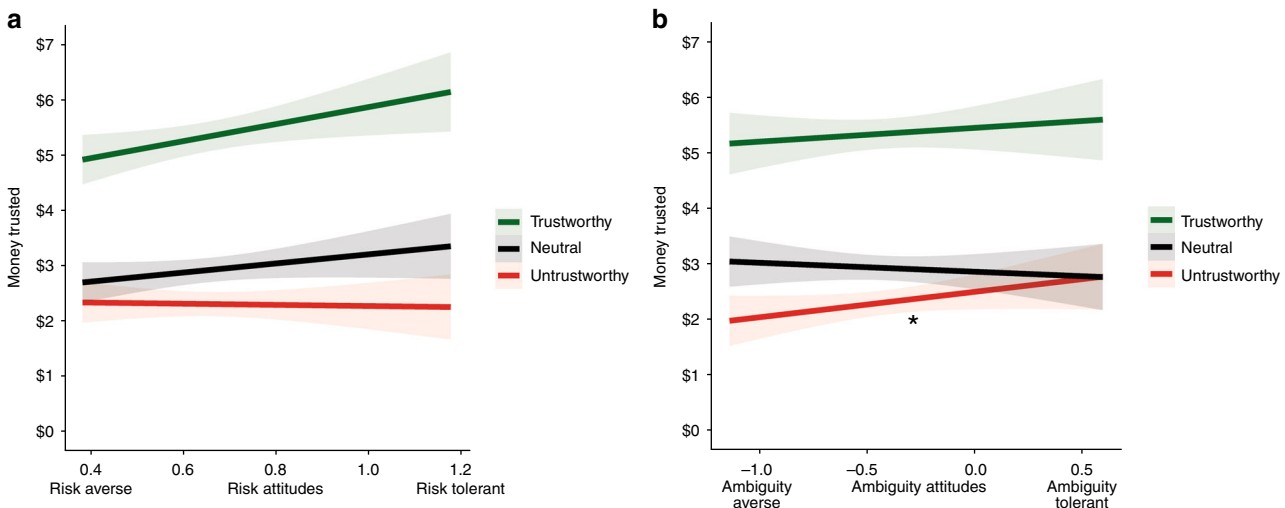

**Fig. 3** Experiment 2 results. **a** Risk attitudes do not modulate trusting behavior. **b** A regression reveals that ambiguity attitudes modulate the amount of money entrusted in the untrustworthy trustee but does not modulate money trusted in the trustworthy or neutral trustee. Ambiguity attitudes are inverted to align on the same scale as risk attitudes for presentation purposes. Fitted regression lines are plotted. Asterisks indicate significant differences (*$p < 0.05$). Error bars reflect 95% CIs

To test our key question of interest—whether tolerance to ambiguity predicts decisions to trust—we conducted a trial-by-trial hierarchical linear regression modeling the amount of money subjects entrusted as a function of trustee type and subjects' ambiguity attitudes (Table 2). Replicating our findings from experiment 1, results reveal that amount of money entrusted was modulated by player type and increasing tolerance to ambiguity: the greater an individual's ability to tolerate ambiguity the more willing they were to entrust their money with the untrustworthy trustee (there was no relationship between ambiguity tolerance and decisions to trust the trustworthy trustee; Fig. 3b). The same analysis for tolerance to risk revealed no effect of risk attitudes on decisions to trust (Supplementary Table 7, Fig. 3a). As in experiment 1, the coefficients for these two predictors (untrustworthy trustee and ambiguity attitudes and untrustworthy trustee and risk attitudes) were significantly different from one another ($z(36) = 2.68$, $p = 0.007$), replicating the finding that ambiguity—but not risk—tolerance influences prosocial behavior (see Supplementary Table 8, which includes risk and ambiguity attitudes in the same model).

Finally, contrary to our prediction, we observed no relationship between ambiguity attitudes and the amount of ambiguity associated with each trustee (Table 3). This may be because our exogenous manipulation of ambiguity (i.e., the number of times a subject was presented with their partner's decision to reciprocate

or defect) was not strong enough. An ambiguous partner's decision was presented up to five times over the task (depending on whether the subject made a choice to trust), which was at half the rate of fully informed partners (whose decision to reciprocate or defect were presented on every trial). It is possible that being exposed to five instances of a partner's moral behavior, especially for an untrustworthy partner, is enough information to disambiguate whether one should trust (or distrust). This would accord with prior work showing that subjects quickly learn to trust after minimal feedback[37], as well as our finding that subjects were only sensitive to the ambiguous manipulation for the trustworthy trustee (Fig. 2c; Table 3).

**Competitive cooperation under uncertainty.** Together, both experiments 1 and 2 illustrate that across different social contexts, ambiguity (but not risk) tolerance influences one's willingness to engage in prosocial behavior. As a final test of the relationship between ambiguity tolerance and prosociality, we examined whether once an individual has reached a state in which they have gathered enough information—and have thus resolved some of the associated ambiguity—there should no longer be an effect of ambiguity tolerance on prosocial behavior. In other words, since aversion to uncertainty is rooted in not knowing the probabilities of outcomes, if there is less uncertainty associated with an

**Table 3 Experiment 2: Exogenous ambiguity levels in the Trust Game**

| DV | Coefficient ($\beta$) | Estimate (SE) | t-value | P-value |
|---|---|---|---|---|
| Money trusted | Intercept | 0.90 (0.11) | 8.18 | <0.001*** |
| | Ambiguity attitude | −0.04 (0.11) | −0.33 | 0.74 |
| | Untrustworthy | −0.22 (0.07) | −3.12 | 0.001** |
| | Trustworthy | 0.36 (0.10) | 3.76 | <0.001*** |
| | Ambiguity attitude × untrustworthy | 0.12 (0.07) | 1.79 | 0.07† |
| | Ambiguity attitude × trustworthy | 0.07 (0.09) | 0.68 | 0.49 |
| | Untrustworthy × ambiguity level | −0.13 (0.09) | −1.32 | 0.18 |
| | Trustworthy × ambiguity level | 0.43 (0.10) | 4.13 | <0.001*** |
| | Untrustworthy × ambiguity attitude × ambiguity level | 0.02 (0.09) | 0.21 | 0.83 |
| | Trustworthy × ambiguity attitude × ambiguity level | 0.04 (0.11) | 0.36 | 0.72 |

Money trusted$_{i,t} = \beta_0 + \beta_1$ ambiguity attitude$_i \times \beta_2$ type of trustee$_{i,t} + \beta_3$ ambiguity level$_{i,t} \times$ type of trustee$_{i,t} + \beta_4$ ambiguity attitude$_i \times$ ambiguity level$_{i,t} \times$ type of trustee$_{i,t} + \varepsilon$
Where ambiguity attitude is indexed by subject ($i$), type of trustee is a categorical variable, and ambiguity level is an indicator variable. Ambiguity attitudes were inverted and standardized before being entered into the regression. Ambiguity level is coded as ambiguous (0) and unambiguous (1) trustees; AIC = 10,275
†$p < 0.1$; **$p < 0.01$; ***$p < 0.001$

outcome and more information about how the social exchange will unfold, ambiguity attitudes should cease to predict decisions to cooperate.

Accordingly, experiment 3 followed the same structure as the previous two experiments with one key difference; in the final phase of the experiment, subjects ($N = 60$) could obtain information about potential partners in a Prisoner's Dilemma (PD; Fig. 4a) by freely sampling from partners' past choices. Thus, in experiment 3, we first estimated subjects' risk ($\alpha$) and ambiguity ($\beta$) attitudes using a version of the gambling task described in experiments 1 and 2 (Fig. 1a; the monetary wins were modified for Mturk, see Supplementary Methods). Subjects then completed three one-shot PD games with no feedback. Finally, subjects completed a task where they could spend time observing how potential partners behaved in previous PD games before deciding whether to cooperate themselves (i.e., the sampling phase)—a within subject design. Unbeknownst to subjects, there were three types of players in the sampling phase: one who had previously cooperated 90% of time; one who defected 90% of the time; and one who cooperated 50% of the time. Once subjects felt they had enough information about a partner, subjects could decide to cooperate or defect. In the PD, mutual cooperation is more beneficial than mutual defection; however, defecting when your partner cooperates reaps the highest payoff. In contrast, cooperating when your partner defects leaves the subject with the lowest possible payoff. Together this task structure allowed us to assess both whether ambiguity tolerance predicts cooperative behavior in yet another social context (the PD), while also testing if the relationship between ambiguity tolerance and cooperation can be abolished if subjects resolve some of the ambiguity endemic to the social exchange through sampling.

Replicating experiments 1 and 2, results reveal that tolerance to ambiguity predicts cooperative, prosocial behavior in the first phase of the PD (Table 4, Fig. 4b), while tolerance to risk did not modulate cooperative behavior (Table 4). As before, the relationship between ambiguity tolerance and cooperation was significantly more predictive than the same relationship with risk (coefficients tested from the regressions in Table 4: $z(59) = 2.14$, $p = 0.03$). We then modeled subjects' choices to cooperate or defect as a function of partner type and the subject's ambiguity attitude during the sampling phase of the experiment. As expected, when the ambiguity about one's partner is sufficiently resolved—in this case through sampling a potential partner's past behavior—there is no longer an observable relationship between ambiguity tolerance and prosocial decisions (Table 5). As before, risk attitudes still did not predict cooperation during the sampling phase of the task (Supplementary Table 9). This suggests that

once ambiguous uncertainty dissipates in a social exchange, an individual's ambiguity attitude ceases to predict decisions to cooperate.

## Discussion

Uncertainty is endemic to decision-making[5,9] and is especially acute during social interactions, as it is difficult to predict how other people will act. Thus, there is a constant demand to estimate the probabilistic distributions associated with another's behavioral repertoire. We examined this social decision space, finding that ambiguity, but not risk, attitudes predict willingness to engage in prosocial behavior. Specifically, the greater an individual's ambiguity tolerance, the more they engage in costly prosocial actions—across multiple different social contexts. We observed this relationship during binary decisions to cooperate (experiments 1 and 3) and granular choices to trust (experiment 2). Finally, we also found that the relationship between ambiguity tolerance and prosocial choice can be abolished if the ambiguity associated with another's actions is sufficiently resolved (experiment 3). Taken together, these results provide converging evidence that one's attitudes toward ambiguous uncertainty are a robust and stable predictor of prosocial behavior, suggesting that a specific cognitive phenotype (e.g., personality variable) underlies—at least in part—the motivation to engage in prosocial action.

Interestingly, the divergent results observed in the aversive and appetitive domain in experiment 2 suggest that individuals hold different perceptions of uncertainty depending on whether the associated outcomes are positive or negative. Trustworthy partners linked with ambiguous outcomes were trusted less than trustworthy partners whose behavior was fully informed. This effect was not observed for untrustworthy partners, such that there was no effect of the ambiguity manipulation on how much (or little) people were willing to entrust to the untrustworthy players. This intimates that while trust is easily eroded under tenuous, uncertain contexts, distrust is less susceptible to the effects of uncertainty—largely because people seem unwilling to trust those who signal that they cannot be trusted. This converges with early research examining negativity biases, which demonstrates that moral reputations are hard to build and easy to lose[38,39]. In other words, the possibility of negative outcomes caused by others looms larger than the possibility of positive ones.

One key feature of ambiguity is that it can be partially resolved by gathering information about how outcomes unfold[40]. As ambiguous probabilities are repeatedly sampled and more information becomes available, the decision space becomes more akin to outcomes with known probabilities, since the underlying probability distributions have been learned. By manipulating how information was obtained in the three different experiments, we

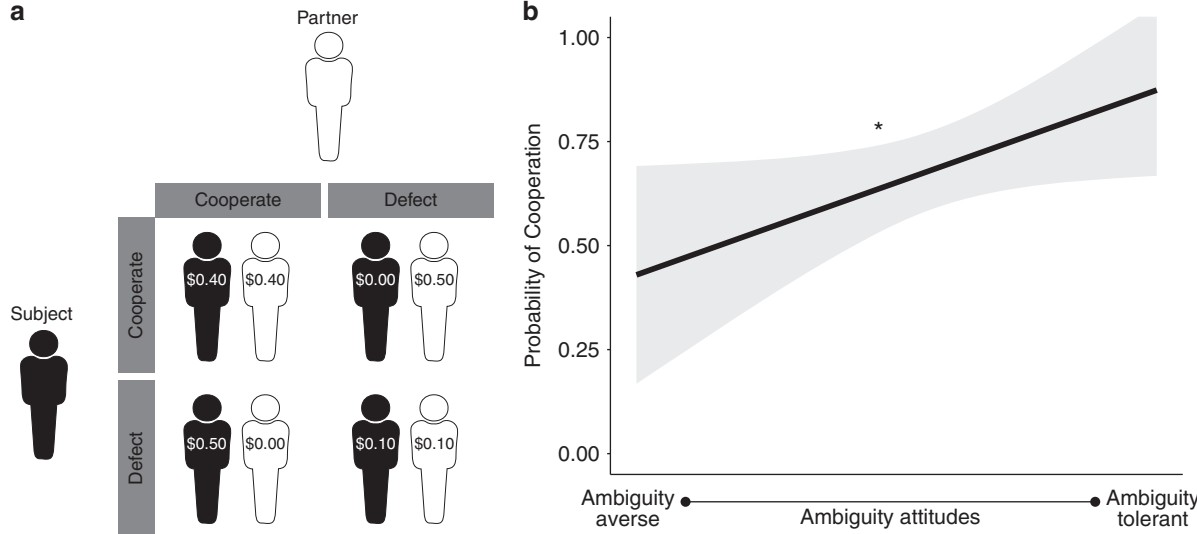

**Fig. 4** Experiment 3 task structure and results. **a** In the prisoner's dilemma, subjects were presented with a matrix of how the payoff structure would work according to their choices and their partner's choices. **b** A regression reveals decision to cooperate was predicted by greater tolerance to ambiguity. As in experiments 1 and 2, ambiguity attitudes are inverted. Fitted regression lines are plotted. Asterisks indicate significant differences (*p < .05). Error bars reflect 95% CIs

**Table 4 Experiment 3: Risk and ambiguity attitudes in the prisoner's dilemma**

| DV | Coefficient (β) | Estimate (SE) | t-value | P-value |
|---|---|---|---|---|
| Cooperation | Intercept | 0.69 (0.04) | 14.67 | <0.001*** |
| | Risk attitudes | −0.05 (0.05) | −1.00 | 0.32 |
| Cooperation | Intercept | 0.68 (0.04) | 14.73 | <0.001*** |
| | Ambiguity attitudes | 0.10 (0.05) | 2.11 | 0.03* |

$\text{Cooperation}_{i,t} = \beta_0 + \beta_1 \text{ risk attitudes}_i + \varepsilon$
$\text{Cooperation}_{i,t} = \beta_0 + \beta_1 \text{ ambiguity attitudes}_i + \varepsilon$
Where risk and ambiguity attitude are indexed by subject (i). Ambiguity attitudes were inverted to align on the same scale as risk attitudes, and risk and ambiguity attitudes were standardized before being entered into the regression. Cooperation is the proportion of cooperative choices taken in the first three trials of the PD where there was not an opportunity to sample
*p < 0.05; ***p < 0.001

explored whether the relationship between ambiguity tolerance and prosocial behavior is modulated by the resolution of ambiguity. Thus, the largest effect was observed in experiment 1, where subjects could freely obtain information about the other players irrespective of whether they themselves cooperated. In other words, for those especially averse to ambiguity, there was no incentive to behave in a costly, prosocial manner in order to reduce ambiguity.

However, even when information about the other players could only be obtained by active and potentially costly decisions to trust (experiment 2), we still found a relationship between ambiguity tolerance and prosocial behavior. This effect was specific to the aversive domain (e.g., the untrustworthy player), which suggests that ambiguity-tolerant individuals have a greater willingness (or optimism) to engage with individuals who signal they are unlikely to reciprocate. When we directly explored the resolution of ambiguity in a within subject design (experiment 3), we found that while ambiguity tolerance predicted prosocial behavior, this relationship could be eliminated if individuals were able to freely access information about how a partner behaved in the past. In other words, even within the same individual, we found evidence that tolerance to ambiguity predicts prosocial behavior when the situation holds great ambiguous uncertainty about what a partner will do, but disappears once this uncertainty is partially resolved.

Finally, expectations about how other people should behave in particular social contexts may further modulate perceptions of uncertainty. In the contexts examined here, there are psychologically distinct social norms that determine how others ought to behave. For example, in the TG, there is a strong social norm to reciprocate: when someone entrusts you with money, the custom is to return this act of kindness. Indeed, research shows that the proportion of money reciprocated increases with the amount of money that is initially entrusted[1,41,42]. The existence of a strong social norm to reciprocate may reduce some of the uncertainty associated with what people will do, which could, in turn, promote greater trust. In contrast, due to the nature of the rules in the PGG and PD (and the fact that the normative behavior is bimodally distributed across decisions to cooperate and defect), the social norm is less clearly defined in these tasks[43]. This lack of social norm may enhance perceptions of uncertainty, since it becomes more difficult to estimate how other players will strategically behave in competitive environments where outcomes can be zero-sum. Further research exploring the ways in which social norms influence ambiguity tolerance will help tease apart how ambiguity is resolved in our social world.

Even though decision-making under risk and uncertainty is one of the more active and interdisciplinary research topics in judgment and decision-making, very little is understood about how these uncertainty constructs influence social decision-making. Here we show that the scope of uncertainty reaches beyond the nonsocial domain, and can powerfully impact people's decisions to cooperate and trust others.

## Methods
**Subjects**. Across three experiments, 250 subjects were recruited. In experiment 1, we ran 110 subjects in laboratories housed at New York University and Universitat Pompeu Fabra. Seven subjects were excluded, resulting in a final sample of 103 subjects (64 female, mean age = 28.1, SD ± 10.10). One subject was excluded for failing to believe that the other players in the PGG task were real, 3 subjects for having an un-estimable risk attitude, and 3 because data was not recorded during the experiment. Subjects were paid $10 (or 10€) for participating, plus an additional bonus of up to $20 (20€) from their decisions in the PGG, and a maximum bonus of $125 (125€) from their decisions in the gambling task. In experiment 2, we recruited 40 subjects at Brown University; 3 were excluded, which resulted in a final sample of 37 subjects (23 female, mean age = 21.4, SD ± 5.01). One subject was excluded for admitting to being aware of the nature of the research, 1 because data was not recorded during the experiment, and 1 for having an un-estimable

**Table 5 Experiment 3: Ambiguity attitudes in the prisoner's dilemma during sampling phase**

| DV | Coefficient ($\beta$) | Estimate (SE) | t-value | P-value |
|---|---|---|---|---|
| Cooperation | Intercept | 0.38 (0.34) | 1.11 | 0.27 |
| | Ambiguity attitudes | −0.03 (0.34) | −0.09 | 0.92 |
| | Defection player | −0.58 (0.46) | −1.40 | 0.16 |
| | Cooperative player | 0.82 (0.43) | 1.90 | 0.06[†] |
| | Ambiguity attitudes × defection player | −0.74 (0.44) | −1.67 | 0.10 |
| | Ambiguity attitudes × cooperative player | 0.36 (0.43) | 0.84 | 0.40 |

Cooperation$_{i,t} = \beta_0 + \beta_1$ ambiguity attitudes$_i \times \beta_2$ types of player$_{i,t} + \varepsilon$
Where ambiguity attitude is indexed by subject ($i$) and type of player is a categorical variable. Ambiguity attitudes were inverted and standardized before being entered into the regression. Cooperation is coded as defection (0) and cooperation (1); AIC = 236.2
[†]$p < 0.1$

risk attitude. Subjects were paid $10 for participation, plus an additional bonus of up to $20 in the TG, and a maximum bonus of $125 in the gambling task. In experiment 3, we ran 100 subjects on Amazon Mechanical Turk[44]. However, 10 subjects' data failed to record, and 30 subjects had an un-estimable risk or ambiguity attitude, leaving a final sample of 60 (19 female, mean age = 33.1, SD ± 8.43). Subjects were paid $1.25 for participating and could earn an additional bonus of up to $6.25 for the gambling task (see Supplementary Methods for more details) and $0.50 for the PD. All experiments complied with ethical regulations, study protocols were approved by each of the local ethics committees, and informed consent was obtained from all participants.

**Gambling task**. Subjects completed 62 trials across two blocks. In each trial, subjects decided whether to take the sure payout of $5 or play the lottery. Each lottery was presented for 6 s, followed by a green dot where the participant had 3.5 s to indicate if she preferred the lottery or the safe bet. Afterwards, a fixation cross was presented for 2 s and the next trial started. At the end of the experiment, the computer randomly selected one trial for payout. If the subject's decision was to gamble, the subject played the lottery using the corresponding bag of chips. Thus, in the case of a 50% chance of winning, the subject would reach into a bag filled with 50 blue chips and 50 red chips. Given that the proportion of the blue and red chips in the ambiguous trial was always 50%, the only difference between risky and ambiguous trials was the amount of information presented to subjects (denoted by the size of the occluder on ambiguous trials). See Supplementary Table 1 for a full list of choice types that were evenly presented across the task. See Supplementary Table 3 for the descriptive statistics of each experiment, and Supplementary Table 4 (and Supplementary Fig. 1) for experiment 1's descriptive statistics broken down by geographical location.

**Public good game**. Subjects completed 20 one-shot trials of the PGG. In each trial, subjects were told that they would interact with different players. Subjects were endowed with $8 on every trial and decided whether they wanted to contribute their money to a common pool. The total amount contributed was doubled and equally split amongst all players. Subjects did not have any time restrictions for responding. After making a decision, subjects received detailed feedback of the contribution of each player and the payoff that all players received on that trial. Decisions to cooperate or freeride from the other players were real choices recorded in previous PGG studies run in our lab.

**Trust game**. Subjects completed 60 trials of TG. Each of the six different trustees was presented 10 times. At the beginning of each trial, subjects were presented with the name and the picture of the trustee and had 3.5 s to decide how much of their $10 they wanted to entrust. Money entrusted was then quadrupled, such that if a subject decided to share $4, the trustee would receive $16. Subjects were subsequently presented with the amount of money the trustee sent back. If subjects did not share any money, the experiment moved forward to the next trial. Decisions to reciprocate or defect were predetermined by the experimenters and followed a specific algorithm (see Supplementary Methods). It was further explained to subjects that sometimes the computer would randomly choose to not show any feedback. Two different feedback rates were created: one that was unambiguous, where feedback was always presented; and one that was ambiguous, such that feedback was only presented half of the time. Each type of trustee was crossed with the three different ambiguity profiles, leading to a total of six different trustees: unambiguous trustworthy; ambiguous trustworthy; unambiguous neutral; ambiguous neutral; unambiguous untrustworthy; and ambiguous untrustworthy. See Supplementary Table 2 for a representation of all the possible ways each trustee could respond.

**Prisoner's dilemma**. In the PD, subjects decided between two options that would affect their own payoff as well as the payoff of their partner. The cooperative option can lead to the most efficient payoff for everyone involved. In our experiment, if both players chose to cooperate, each player would receive $0.40. However, if one player chose to cooperate but the other did not, the defecting partner would receive

($0.50) and the cooperating partner would receive nothing ($0). If both players chose to defect, each player only received $0.10 (Fig. 4a). To ensure that subjects understood the rules of the game, correct responses were enforced in the pre-task comprehension test. Once the sampling phase of the game started, subjects could freely sample past decisions that their partner made with other players in previous games. When enough information was gleaned, participants could choose to cooperate or defect. Unbeknownst to subjects, there was a limit (i.e., up to 50 times) on sampling a partner's past choices. This limit was never reached.

**Analysis**. Gilboa and Schmeidler's maxmin utility model provides a simple and widely used model to successfully estimate risk and ambiguity attitudes[11,12,31]. We employed the following utility function, which takes into account the effect of ambiguity on the perceived winning probability as

$$SV(p, A, v) = \left( p - \beta \times \frac{A}{2} \right) \times v^{\alpha}$$

where for each trial, SV is calculated as a function of the lottery's objective winning probability ($p$), level of ambiguity ($A$), and monetary value ($v$), accounting for each individual's risk ($\alpha$) and ambiguity ($\beta$) attitudes, which are obtained from the behavioral fit of the model. These attitudes were derived by fitting choice data using the maximum likelihood with the following probabilistic choice function:

$$P(\text{chose lottery}) = \frac{1}{1 + e^{\gamma(SV_F - SV_V)}}$$

where $SV_F$ and $SV_V$ are the subjective values of the fixed ($F$) and variable ($V$) options, respectively, and $\gamma$ is the slope of the logistic function, which is a participant-specific parameter. These individual risk and ambiguity attitudes were then used as predictor variables in hierarchical regressions. However, to remove biases when comparing effects of two variables against one another[45–47], we standardized ambiguity and risk attitudes before entering them into the regressions. All reported trial-by-trial regressions were run using Matlab's fitlme function, fitting fixed and random (subject-specific) slopes for each variable, as well as random intercepts for each participant[48].

**Data availability**. All data are available on Open Science Framework (OSF) through DOI 10.17605/OSF.IO/AHYQJ. Codes used to generate the analysis are available upon request.

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

## Acknowledgements

We thank Julian Wills and Augustus Baker for helping collect data in experiment 1 and Joachim Krueger for his thoughtful and insightful comments on our manuscript. M.L.V. was supported by a grant from the Spanish Government (BES-2015-071581).

## Author contributions

M.L.V. and O.F.H. developed and designed the experiments. M.L.V. conducted data collection. M.L.V. and O.F.H. performed the statistical analyses. M.L.V. and O.F.H. wrote the paper.

## Additional information

**Competing interests:** The authors declare no competing interests.

