## [Peer Review File · Nature Communications]

Reviewers' comments:

Reviewer #1 (Remarks to the Author):

Tolerance to uncertainty predicts prosocial behavior

This paper offers an interesting and clean distinction between risk and ambiguity. It is confusing strange that the title talks about tolerance to uncertainty, as the author/s even in the abstract make the distinction between ambiguous and risky uncertainty. While the distinction is appreciated, I do not think that the specific tests allows the researchers to conclude that tolerance to ambiguous uncertainty is more important than risky uncertainty. Study 1 simply shows that risk x trial has a t-value of -1.42, while the ambiguity x trial has a t-value of -2.66. So, this should leave one, statistically, unconvinced. The distinction is not further tested critically in Studies 2 and 3. In fact, later the author/s note that they are simply interested in whether tolerance to ambiguity predictions to trust, which does not seem a very ambitious goal. It would be more ambitious if they show that it is only the ambiguity above and beyond risk that accounts for the tendencies to trust and prosociality. That test is never provided. But even if that test would be significant across studies (which I doubt) then, in my view, the magnitude of the empirical contribution would merit publication in Nature Communication. It is a nice to know, but still an incremental contribution.

Other concerns are why the authors found that ambiguity tolerance is relevant to Untrustworthy Trustees (and not to Trustworthy Trustees) In Study 2, while Study 3 reveals still different patterns. As to Study 3, people would like to see a significant effect (that the relationship between ambiguity tolerance and prosocial behavior can be abolished) as expressed by a significant statistical interaction of ambiguity tolerance and a variable that gets at degree of abolishment. A study with another design would be required. Again, the ultimate test of the key role of ambiguity tolerance (independent of risk) has yet to be provided.

Reviewer #2 (Remarks to the Author):

This paper examines the relationship between ambiguity attitudes and prosocial behavior. In the first two experiments, the authors use a gambling task with monetary rewards, to characterize individual attitudes towards risk and ambiguity, and either a public-goods game (PGG) or a trust game (TG) to characterize prosocial behavior. In one experiment, the authors report that ambiguity, but not risk, attitudes predicted contributions to a common pool in the PGG, and that these attitudes became less predictive with time, after subjects received more and more feedback, which reduced the level of ambiguity. In another study, subjects played the role of the investor in a TG, against trustworthy, untrustworthy and neutral trustees. Subjects received full feedback about the decisions of some trustees, while for other trustees, only partial feedback was provided, resulting in a 3 (trustworthiness) by 2 (ambiguity level) design. The authors report decreased trust in the ambiguous trustworthy trustee, compared to the unambiguous one, an opposite effect for the neutral trustee, and no effect for the untrustworthy one. Tolerance to ambiguity in the gambling task predicted the amount of money transferred to the untrustworthy trustee, but not to the other trustees. In the final experiment, the authors report correlation between ambiguity aversion, this time measured with a questionnaire, and the time spent sampling past behavior of an opponent in the Prisoner's Dilemma (PD) game. The authors reason that the sampling procedure allowed subjects to resolve the ambiguity, which explains why subsequent behavior on the game itself was not predicted by ambiguity attitudes.

This is an interesting paper, which asks an intriguing and unexplored question – what is the role of individual uncertainty attitudes in social behavior. A strength of the paper is the distinction between risk and ambiguity, and the use of a separate, non-social, task to estimate risk and ambiguity

attitudes. The study design is elegant, and the paper is clearly written. I am worried, however, that the results are quite weak, and have some questions regarding the design, analysis and results.

- Risk and ambiguity attitudes are quantified with a choice task in experiments 1 and 2, but in experiment 3 subjects instead fill out the Multidimensional Attitude Towards Ambiguity – why? What is the relationship between these two measures of ambiguity attitudes?

- The results of experiment 1 were a bit confusing to me. The authors report a significant effect of ambiguity attitudes, and a significant interaction between ambiguity attitude and trial number, both interesting findings. The authors reason that tolerance to ambiguity leads to higher willingness to contribute, and that as ambiguity is reduced with time individual differences in ambiguity attitudes should play a smaller role in the decision to contribute. From the visualization of the effect in Figure 1D, however, it seems that those individuals who were tolerant to ambiguity are the ones that changed their behavior along the experiment, and contributed less with time. Conversely, ambiguity-averse subjects did not seem to change their behavior – this seems at odd with the authors' interpretation of the results.

- The results of experiment 2 are also not completely clear and seem rather weak. Were the post-hoc tests corrected for multiple comparisons? Ambiguity reduced the transfer to the trustworthy trustee, but not to the untrustworthy trustee, which the authors interpret as reflecting different perceptions of uncertainty in the positive and negative domains. If that's the case, what is the explanation for the opposite effect observed for the neutral trustee (increased transfers in the ambiguous condition)? In addition, the use of two models was confusing to me – why not just use the full model? Was there a main effect of ambiguity level in either of these models? I also did not understand the authors' explanation for the lack of correlation between ambiguity attitudes and the effect of ambiguity level – isn't the exogenous manipulation of ambiguity exactly what you would expect to interact with individual ambiguity attitudes?

- More generally, all of my concerns would be alleviated if the results were stronger. A potential remedy would be to replicate the findings in a new sample. The numbers of subjects in experiments 1 and 2 are moderate (38 and 37), and a replication should be relatively easy, and will substantially strengthen the manuscript. Experiment 3 includes 99 subjects, so a potential way to go would be to divide the data into two and see whether the effect is replicated in each half.

Reviewer 1:

R1 Comment 1: This paper offers an interesting and clean distinction between risk and ambiguity. It is confusing strange that the title talks about tolerance to uncertainty, as the author/s even in the abstract make the distinction between ambiguous and risky uncertainty. While the distinction is appreciated, I do not think that the specific tests allows the researchers to conclude that tolerance to ambiguous uncertainty is more important than risky uncertainty. Study 1 simply shows that risk x trial has a t-value of -1.42, while the ambiguity x trial has a t-value of -2.66. So, this should leave one, statistically, unconvinced. The distinction is not further tested critically in Studies 2 and 3. In fact, later the author/s note that they are simply interested in whether tolerance to ambiguity predicts to trust, which does not seem a very ambitious goal. It would be more ambitious if they show that it is only the ambiguity above and beyond risk that accounts for the tendencies to trust and prosociality. That test is never provided. But even if that test would be significant across studies (which I doubt) then, in my view, the magnitude of the empirical contribution would merit publication in Nature Communication. It is a nice to know, but still an incremental contribution.

Response 1: The Reviewer brings up a couple of important points, and we address each in turn. First, there is the issue that the original title only referenced uncertainty and did not specifically link ambiguity with social choice. We have now edited the title to better reflect the findings: "Tolerance to ambiguous uncertainty predicts prosocial behavior".

Second, the Reviewer mentions that we do not directly test whether tolerance to ambiguous uncertainty is more predictive than risky uncertainty. This is a good point, and one we should have included in the original submission. Determining whether one variable is more significant than another can be done in a number of ways. First, if the independent variables are standardized before being used in the regression, then each predictor is on the same scale and their coefficients can be directly compared, such that the coefficient with the highest absolute value reflects the largest effect. Accordingly, we have now standardized all our variables before entering them into each regression. Across all experiments, the coefficient for ambiguity attitudes—but not risk—is significant. For example, as you can see in Table 1 reported in the manuscript (page 6), the coefficient for the ambiguity attitude (and the ambiguity attitude x trial) are larger than the respective risk coefficients.

Another method is to run separate regressions, one for risk and one for ambiguity and then formally compare the strength of the models (through AIC, where lower AIC reflects better model fits) and/or by directly testing the significance of the beta coefficients from each regression (Clogg, Petkova, & Haritou, 1995). Thus, if i) the risk coefficient is not predictive of prosocial choice (described in the above paragraph), ii) the risk model has a higher AIC, and iii) the risk coefficient is significantly worse at predicting the outcome when directly compared to the ambiguity coefficient, it can be taken as good evidence that ambiguous uncertainty is more predictive than risky uncertainty. We ran these tests for all experiments (including the new replication studies). In every experiment, ambiguity attitudes are predictive of prosocial decisions while risk attitudes are not. That is, the coefficients for Ambiguity attitudes are significantly different from risk attitudes ($P_s < 0.05$) and the AIC is always lower for the ambiguity model compared to the risk model.

To illustrate this point, below we copy the data from Experiment 1. As you can see, ambiguity attitudes and the interaction between Ambiguity and Trial predict cooperative behavior (Table 1), while risk attitudes and the interaction between Risk and Trial do not (Table 2). When we formally test the predictive strength of these coefficients against one another we find evidence that ambiguity attitudes are significantly more predictive of cooperation than risk attitudes: $z(102) = 2.14$, $p\text{-value} = 0.03$ (a similar result is found if we use the coefficients from the complete model [Table 1 in the manuscript, which includes both ambiguity and risk in the same model] rather than the simple regressions). Converging with this, the AIC for the ambiguity model reported in Table 1 is lower than the AIC of the risk model reported in Table 2. Together, these results increase our confidence that ambiguity attitudes—and not risk attitudes—solely predict cooperative behavior.

NB: Based on the feedback from Reviewer 2, we ran a replication study of Study 1. Thus, in the revised submission, we combine the data from Study 1 with the replication study (total $N = 103$) and report the findings together.

Table 1 | Experiment 1 (N=103):

$$\text{Cooperation}_{i,t} = \beta_0 + \beta_1 \text{Risk Attitude}_i \times \beta_2 \text{Trial Number}_{i,t} + \beta_3 \text{Ambiguity Attitude}_i \times \beta_4 \text{Trial Number}_{i,t} + \varepsilon$$

DV	Coefficient (β)	Estimate (SE)	t-value	P value
Cooperation	Intercept	-0.15 (.19)	-0.80	0.42
	Trial	-0.08 (.01)	-6.44	<0.001***
	Ambiguity Attitude	0.66 (.21)	3.14	<0.001**
	Ambiguity Attitude X Trial	-0.03 (.01)	-1.86	0.06†

Note. Where Ambiguity Attitude is indexed by subject (i) and Trial Number is indexed by subject and trial (i, t). Ambiguity Attitudes were inverted to align on the same scale as Risk Attitudes. Cooperation is coded as defect (0) and cooperate (1). AIC=2013.2.

Table 2 | Experiment 1 (N=103):

$$\text{Cooperation}_{i,t} = \beta_0 + \beta_1 \text{Risk Attitude}_i \times \beta_2 \text{Trial Number}_{i,t} + \beta_3 \text{Ambiguity Attitude}_i \times \beta_4 \text{Trial Number}_{i,t} + \varepsilon$$

DV	Coefficient (β)	Estimate (SE)	t-value	P value
Cooperation	Intercept	-0.17 (.19)	-0.86	0.38
	Trial	-0.08 (.01)	-6.25	<0.001***
	Risk Attitude	0.04 (.20)	0.19	0.85
	Risk Attitude X Trial	-0.02 (.01)	-1.23	0.22

Note. Where Risk Attitude is indexed by subject (i) and Trial Number is indexed by subject and trial (i, t). Ambiguity Attitudes were inverted to align on the same scale as Risk Attitudes. Cooperation is coded as defect (0) and cooperate (1). AIC=2072.1.

R1 Comment 2: Other concerns are why the authors found that ambiguity tolerance is relevant to Untrustworthy Trustees (and not to Trustworthy Trustees) In Study 2, while Study 3 reveals still different patterns. As to Study 3, people would like to see a significant effect (that the relationship between ambiguity tolerance and prosocial behavior can be abolished) as expressed by a significant statistical interaction of ambiguity tolerance and a variable that gets at degree of abolishment. A study with another design would be required. Again, the ultimate test of the key role of ambiguity tolerance (independent of risk) has yet to be provided.

Response: We first address the point that the behavioral patterns in Experiments 2-3 are different. We would like to highlight that Experiment 2 and Experiment 3 (in the original submission) were psychologically and structurally different, which makes it theoretically difficult to directly compare one with the other. To expound on this, in the original submission, Experiment 2 sought to replicate the findings from Experiment 1 in a new context (Trust), while Experiment 3 sought to abolish the relationship between ambiguity attitudes and prosocial choices. In Experiment 2 we found that ambiguity attitudes (but not risk attitudes) predicted decisions to trust another player, especially for untrustworthy partners. That we observed a relationship between ambiguity attitudes and untrustworthy partners but not a relationship between ambiguity attitudes and trustworthy partners is likely to be a function of the hazard that is uniquely associated with trusting someone who fails to reciprocate: it is worse to trust and have money lost and accrued a broken relationship, then to trust and have money gained and an intact relationship (De Dreu & McCusker, 1997). In the original submission, the goal of Experiment 3 was to eradicate this relationship altogether by allowing subjects to gather information about the moral character of potential partners before engaging with them. In line with our hypothesis, when subjects were able to observe the past behavior of these potential partners, the relationship between an uncooperative partner and a subject's ambiguity attitudes was entirely abolished ($P=0.98$). In other words, being able to accrue information about your partner's past behavior effectively eliminated the relationship between one's tolerance to ambiguity and cooperative behavior because there no longer is a need to actively take the risk of trusting someone to determine whether they are in fact trustworthy.

The Reviewer suggested, however, that we run an additional experiment with a new design to first illustrate a significant effect between ambiguity attitudes and prosocial behavior, and then abolish the effect within the same subjects. We agree that this would provide strong evidence, and have now run an additional experiment that does just that. In this new Experiment 3 (which has replaced the original third experiment), we followed the same structure as the previous two experiments with one key difference; in the final phase of the experiment subjects could obtain information about potential partners in a Prisoner's Dilemma (PD) by freely sampling from partners' past choices. Thus, we first estimated subjects' risk and ambiguity attitudes using the gambling task. Subjects then completed three one-shot PD games with no feedback, before partaking in a task where they could spend time observing how potential partners behaved in previous PD games before choosing to cooperate (i.e., the sampling phase). This within subject task structure allowed us to assess both whether ambiguity tolerance predicts cooperative behavior in yet another social context (the PD), while also testing if the relationship between ambiguity tolerance and cooperation can be abolished if subjects resolve some of the ambiguity endemic to the social exchange through sampling. Replicating Experiments 1 and 2, results reveal that tolerance to ambiguity predicts cooperative, prosocial behavior in the PD, while tolerance to risk did not. When we modeled subjects' choices to cooperate or defect as a function of partner type and the subject's ambiguity attitude during the sampling phase of the experiment, we found that when ambiguity about one's partner is sufficiently resolved—in this case through sampling a potential partner's past behavior—there is no longer an observable relationship between ambiguity tolerance and prosocial decisions. We hope that these additional results

requested by the Reviewer and found in a within subjects sample helps to attenuate any concerns about the relationship between ambiguity and prosocial choice.

Reviewer 2:

R2 Comment 1: This is an interesting paper, which asks an intriguing and unexplored question – what is the role of individual uncertainty attitudes in social behavior. A strength of the paper is the distinction between risk and ambiguity, and the use of a separate, non-social, task to estimate risk and ambiguity attitudes. The study design is elegant, and the paper is clearly written. I am worried, however, that the results are quite weak, and have some questions regarding the design, analysis and results. Risk and ambiguity attitudes are quantified with a choice task in experiments 1 and 2, but in experiment 3 subjects instead fill out the Multidimensional Attitude Towards Ambiguity – why? What is the relationship between these two measures of ambiguity attitudes?

Response: Thank you for your kind words. We understand the Reviewer's concerns about the i) strength of the findings and ii) whether there is a relationship between the estimated uncertainty attitudes elicited from the gambling task in Experiments 1 and 2 and the Multidimensional Attitude Towards Ambiguity Scale (i.e., a survey) used to measure ambiguity attitudes in the third experiment. First, in regards to the strength of the findings, we have now run two additional replication studies, and these data support the original findings (details on these experiments can be found below, as well as in response to R1's second comment). Second, we originally used a survey measurement for purely technical reasons: while the first two experiments were run in the lab, the third study was run on Mturk. At the time, we made the (flawed) assumption that the MAAS would capture ambiguity tolerance in a similar way to our gambling task and ambiguity estimation procedure, however, we never checked this. Indeed, after receiving this feedback, we ran a pilot study to determine whether MAAS scores correlated with ambiguity estimates from the gambling task. Unfortunately, we did not observe a correlation between these two measures, which led us to remove the third experiment in the original submission.

Taking the advice of the Reviewer, we ran an additional experiment with the same gambling task used in Experiments 1-2 (but changed the monetary values to be Mturk appropriate). This not only enables continuity across all Experiments, but also allows us to directly compare risk and ambiguity attitudes in the same sample (something we were unable to do with the MAAS as it only captures ambiguity and not risk attitudes). Our findings accord with the other experiments: tolerance to ambiguity predicts cooperative, prosocial behavior in the PD ($P=.03$, see Table 4, Fig 4B), while tolerance to risk did not modulate cooperative behavior ($P=0.32$, see Table 4, Fig S3).

R2 Comment 2: The results of experiment 1 were a bit confusing to me. The authors report a significant effect of ambiguity attitudes, and a significant interaction between ambiguity attitude and trial number, both interesting findings. The authors reason that tolerance to ambiguity leads to higher willingness to contribute, and that as ambiguity is reduced with time individual differences in ambiguity attitudes should play a smaller role in the decision to contribute. From the visualization of the effect in Figure 1D, however, it seems that those individuals who were tolerant to ambiguity are the ones that changed their behavior along the experiment, and contributed less with time. Conversely, ambiguity-averse subjects did

not seem to change their behavior – this seems at odd with the authors’ interpretation of the results.

Response: The Reviewer points out that the figures from Experiment 1 may lead a reader to believe that the effect is driven by ambiguity tolerance and not aversion. We apologize that this figure may have been misleading. In order to visualize the results (it is always difficult to plot 3 variables on one graph!), we median split ambiguity attitudes, characterizing subjects as either ambiguity tolerant or ambiguity averse. In reality, however, ambiguity attitudes are on a continuous scale and the results from the regression reflect this (e.g., tolerance to ambiguity and aversion to ambiguity are on opposite sides of the same continuum). When we plot the results using this median split method, it appears as if the results are a function of those who are ambiguity tolerant and not ambiguity averse, when in fact ambiguity attitudes is continuous and includes both tolerance and aversion. To minimize confusion surrounding the graphs, we have replotted the data using a 3D approach. This allows us to graphically illustrate that, as the Reviewer correctly states above, tolerance to ambiguity leads to a higher willingness to contribute early on in the PGG, and that as ambiguity is reduced with time (trials), individual differences in ambiguity attitudes play a smaller role in the decision to contribute. In other words, those who are tolerant to ambiguity are more willing at the beginning of the experiment to cooperate when ambiguous uncertainty is greatest, and this tendency to cooperate decreases over time. We copy this graph below (which has also replaced the original graphs from Figure 1, and which can be found on page 6 of the manuscript, Fig 1C-D).

R2 Comment 3: The results of experiment 2 are also not completely clear and seem rather weak. Were the post-hoc tests corrected for multiple comparisons? Ambiguity reduced the transfer to the trustworthy trustee, but not to the untrustworthy trustee, which the authors interpret as reflecting different perceptions of uncertainty in the positive and negative domains. If that’s the case, what is the explanation for the opposite effect observed for the neutral trustee (increased transfers in the ambiguous condition)?

Response: The Reviewer is right and we should have corrected for multiple comparisons in this analysis. We have revised the manuscript accordingly (by Bonferroni correcting the post-hoc tests) and once done, the Neutral trustee finding is no longer significant. The only significant result that passes Bonferroni correction is the difference between the Ambiguous and fully informed (Unambiguous) trustworthy partners. We have now revised the manuscript to reflect these new results and have removed the finding about the neutral trustee.

R2 Comment 4: In addition, the use of two models was confusing to me – why not just use the full model? Was there a main effect of ambiguity level in either of these models?

Response: We decided to split the analysis in two different models for a matter of simplicity. The effect still holds if we run the analysis with Risk and Ambiguity Attitudes together as predictors in the same model, as can be seen in the table below. We have now included this model in the supplement (Table S9) but would also be happy to replace the simple regressions with the complete model found below if the Reviewer thinks it would strengthen the manuscript.

DV	Coefficient (β)	Estimate (SE)	t-value	P value
Money Trusted	Intercept	0.57 (.35)	1.62	0.10
	Ambiguity Attitude	-0.03 (.27)	-0.11	0.91
	Risk Attitude	0.48 (.52)	0.91	0.35
	Untrustworthy	0.008 (.19)	-0.04	0.96
	Trustworthy	0.77 (.30)	2.51	0.01*
	Ambiguity Attitude x Untrustworthy	0.30 (.14)	2.06	0.03*
	Ambiguity Attitude x Trustworthy	0.21 (.23)	0.91	0.36
	Risk Attitude x Untrustworthy	-0.21 (.28)	-0.76	0.44
	Risk Attitude x Trustworthy	-0.06 (.44)	-0.15	0.87

Note. Where Ambiguity and Risk Attitude are indexed by subject (*i*) and Type of Trustee is a categorical variable, such that the neutral Trustee serves as the reference category. Variables were standardized before being entered into the regression and Ambiguity Attitudes were inverted to align on the same scale as Risk Attitudes. **p*<0.05

R2 Comment 4: I also did not understand the authors’ explanation for the lack of correlation between ambiguity attitudes and the effect of ambiguity level – isn’t the exogenous manipulation of ambiguity exactly what you would expect to interact with individual ambiguity attitudes?

Response: We apologize that this was not clear in the original submission. We were surprised at the lack of correlation between ambiguity attitudes and our exogenous manipulation of ambiguous uncertainty. We reasoned that it is possible that the failure to observe a relationship between these two variables may be due to the fact that our manipulation was not strong enough. In our task, once a decision to trust was made, feedback from partners who were ambiguous were shown at half the rate (5 trials) as partners who were unambiguous (all 10 trials), with the idea being that less information about a partner’s reciprocation or defection rate makes the social exchange more uncertain. However, it is possible that being exposed to 5 pieces of information about a person’s moral character (through feedback about a partner’s decision to reciprocate or defect) is enough

evidence to disambiguate whether it is a good idea to trust, even for those who are ambiguity tolerant. Indeed, within the trust literature, subjects typically learn whether to trust (or distrust) their partner after reading a short description of their partner's moral character (Delgado, Frank, & Phelps, 2005). To put it another way, it is possible that our exogenous manipulation of ambiguity was not ambiguous enough. We have now clarified this point in the manuscript, page 10.

R2 Comment 5: More generally, all of my concerns would be alleviated if the results were stronger. A potential remedy would be to replicate the findings in a new sample. The numbers of subjects in experiments 1 and 2 are moderate (38 and 37), and a replication should be relatively easy, and will substantially strengthen the manuscript. Experiment 3 includes 99 subjects, so a potential way to go would be to divide the data into two and see whether the effect is replicated in each half.

Response: We agree with the reviewer that our results would be stronger with a replication in a new sample. Thus, we have run two more studies, resulting in the recruitment of 125 additional subjects. Specifically, we ran Experiment 1 on 65 additional subjects and have combined this additional data with the existing data, such that Experiment 1 now includes 103 subjects. As expected, the results hold and are even stronger than before ($p < 0.001$ for ambiguity attitudes predicting cooperation; Table 1, page 6). In addition, based on your feedback as well as the feedback from Reviewer 1, we re-ran Experiment 3 using the gambling task (rather than a survey—rationale described above), and found that using a within subjects design, we can first establish the relationship between ambiguity tolerance and increased cooperation (Table 4 in manuscript, page 14), and then abolish this relationship when subjects are able to sample from their partner's past behavior (Table 5 in manuscript, page 14). With these additional experiments and subjects, we are even more confident in our results and are hopeful that the Reviewer is as well.

REVIEWERS' COMMENTS:

Reviewer #1 (Remarks to the Author):

I have read the revision, and I am pleased to note that all my concerns have been successfully reduced so that I have sufficient confidence in the overall conclusions of this paper. Ambiguity matters more than risk in uncertainty in the situations examined. In my view, the authors have done a good job at collecting more data, and performing statistical analyses that, taken together, provide a more convincing test that ambiguity matters more than risk.

Also, I like the authors' reasoning why ambiguity should matter so much in the social (dyadic) world. I wonder, though, why risk do not seem to matter at all, at least in the games studied. Potential explanations for the latter non-finding deserve a bit more attention, in my view.

Reviewer #2 (Remarks to the Author):

The authors have done an impressive job in responding to reviewers' comments. I especially commend them for collecting the additional data, which substantially strengthen their arguments.

I have a couple of remaining minor comments:

- I think my comment about the dynamic changes in experiment 1 was not clear. The authors now provide a new Figure (1D), which is very nice, but, unless I'm missing something, shows the same thing that was shown before. That is, the subjects who changed their behavior along the experiment were those that were more tolerant to ambiguity. Subjects who were relatively ambiguity averse, did not cooperate much in the beginning, and continued to behave in a similar manner, even when ambiguity was reduced. Subjects who were relatively ambiguity tolerant, cooperated more at the beginning, but reduced their cooperation with the resolving of ambiguity. Was this an adaptive change in the context of this experiment? In other words, knowing the behavior of other players, was it more beneficial to cooperate less?
- A related question is about Experiment 3: the authors report that once ambiguity was resolved, ambiguity attitudes no longer contributed to cooperation – was this change in the same direction as in Experiment 1, i.e. did ambiguity-tolerant subjects cease to cooperate? And was it an adaptive change in the context of Experiment 3?

Reviewer #1

I have read the revision, and I am pleased to note that all my concerns have been successfully reduced so that I have sufficient confidence in the overall conclusions of this paper. Ambiguity matters more than risk in uncertainty in the situations examined. In my view, the authors have done a good job at collecting more data, and performing statistical analyses that, taken together, provide a more convincing test that ambiguity matters more than risk. Also, I like the authors' reasoning why ambiguity should matter so much in the social (dyadic) world. I wonder, though, why risk do not seem to matter at all, at least in the games studied. Potential explanations for the latter non-finding deserve a bit more attention, in my view.

Response 1: We are delighted to hear that the Reviewer is satisfied with our revision. As to the Reviewer's question regarding our reasoning for why ambiguity should matter so much in the social world: Humans are capable of being uncertain about everything they attempt to predict, be it features of stimuli, rewards or punishments that can be obtained, actions to be selected, and the manner in which those actions will be executed. From this perspective, social environments are particularly rife with uncertainty. When interacting with others, each of our uncertainties about our own future states and actions is compounded by the fact that we are often also uncertain about who these individuals are (e.g., their motives) and how they might choose to act. This is akin to being in a state of ambiguous uncertainty, where the probabilities of any outcome are not known. It is for this reason that ambiguous uncertainty (and not risky uncertainty) is likely to be more predictive of social decision-making. We have now added in two sentences clarifying this point (pg 3).

Reviewer #2:

The authors have done an impressive job in responding to reviewers' comments. I especially commend them for collecting the additional data, which substantially strengthen their arguments.

I have a couple of remaining minor comments:

- I think my comment about the dynamic changes in experiment 1 was not clear. The authors now provide a new Figure (1D), which is very nice, but, unless I'm missing something, shows the same thing that was shown before. That is, the subjects who changed their behavior along the experiment were those that were more tolerant to ambiguity. Subjects who were relatively ambiguity averse, did not cooperate much in the beginning, and continued to behave in a similar manner, even when ambiguity was reduced. Subjects who were relatively ambiguity tolerant, cooperated more at the beginning, but reduced their cooperation with the resolving of ambiguity. Was this an adaptive change in the context of this experiment? In other words, knowing the behavior of other players, was it more beneficial to cooperate less?

Response 2: Defecting in a public goods game (PGG) is always the most efficient behavior in terms of maximizing individual monetary payoffs. In line with this, a frequent finding in the PGG literature is that people decrease their cooperation over time (Camerer & Fehr, 2004). One way to conceptualize our findings is that in standard versions of the PGG (such as the one employed in Experiment 1) it is 'adaptive' to cooperate less (in so much as that it garners

the individual the most money). It is the people who are more ambiguity tolerant that exhibit this 'adaptive' tendency. It could also be argued, however, that those who are ambiguity intolerant are behaving adaptively from the beginning of the task, since they maintain low levels of cooperation throughout.

- A related question is about Experiment 3: the authors report that once ambiguity was resolved, ambiguity attitudes no longer contributed to cooperation – was this change in the same direction as in Experiment 1, i.e. did ambiguity-tolerant subjects cease to cooperate? And was it an adaptive change in the context of Experiment 3?

Response 3: Although Experiment 1 and 3 share some of the same features (which might make it appealing to compare the results across the two experiments), there are fundamental differences that precludes this possibility. For example, in Experiment 1 participants had no prior knowledge about the other players' past actions before making their decision, while in Experiment 3 participants could freely sample information from their partners' past choices before deciding which strategy to follow. Moreover, unlike in Experiment 1, Experiment 3 presented participants with three types of individuals that varied in cooperation rate. Participants cooperated more when paired with a cooperative player (Supplementary Table 9), and this effect was not modulated by ambiguity attitudes (there was no interaction between Ambiguity Attitudes and Type of Player; Table 5). In other words, ambiguity tolerant participants did not cease to cooperate, rather they adaptively cooperated in a cooperative environment to the same degree that ambiguity averse participants did. This is likely a function of the fact that once ambiguity is sufficiently resolved (through sampling past behavior), ambiguity attitudes stop predicting prosocial behavior.